# Online Bipartite Matching with Advice: Tight Robustness-Consistency Tradeoffs for the Two-Stage Model

**Billy Jin**
Cornell University
bzj3@cornell.edu

**Will Ma**
Columbia University
wm2428@gsb.columbia.edu

## Abstract

We study the two-stage vertex-weighted online bipartite matching problem of Feng, Niazadeh, and Saberi (SODA '21) in a setting where the algorithm has access to a suggested matching that is recommended in the first stage. We evaluate an algorithm by its robustness $R$, which is its performance relative to that of the optimal offline matching, and its consistency $C$, which is its performance when the advice or the prediction given is correct. We characterize for this problem the Pareto-efficient frontier between robustness and consistency, which is rare in the literature on advice-augmented algorithms, yet necessary for quantifying such an algorithm to be optimal. Specifically, we propose an algorithm that is $R$-robust and $C$-consistent for any $(R, C)$ with $0 \leq R \leq \frac{3}{4}$ and $\sqrt{1-R} + \sqrt{1-C} = 1$, and prove that no other algorithm can achieve a better tradeoff.

## 1 Introduction

Online bipartite matching is a fundamental model used to assign incoming requests to servers, incoming workers to jobs, incoming impressions to online advertisers, etc. Traditionally, it has been studied in either the *adversarial* setting, where nothing is assumed about the future; or the *stochastic* setting, where future arrivals are assumed to follow a distributional model. However, algorithms for the former tend to be overly conservative in their decisions; whereas algorithms for the latter can perform terribly when the assumed model is wrong. Therefore, a recently budding literature has studied *advice-augmented* online algorithms, which use a machine-learned prediction to refine their decisions, but do not perform as terribly when the prediction is wrong.

In many problems including online matching, there is an inherent tradeoff between an algorithm's performance when the prediction is correct, called its *consistency*, and the algorithm's performance under the worst-case future, called its *robustness*. This is because a high consistency can only be achieved by optimizing as if the predicted future was correct, resulting in a decision that can be terrible in the worst case. Conversely, a high robustness requires hedging against all possible futures, which does not sufficiently prioritize the one predicted to come true.

In this paper we characterize a *tight* robustness-consistency tradeoff for online bipartite matching, by adding advice to the model of Feng et al. [1] where the online vertices arrive in two stages (but many online vertices can arrive simultaneously). That is, we derive a continuum of algorithms for two-stage online bipartite matching that define the *Pareto-efficient frontier* of the tradeoff between performance when a prediction is right vs. performance when it is wrong. To our knowledge, Pareto-efficiency results are rare in the literature on advice-augmented online algorithms, especially when it comes to online matching (as we review in Subsection 1.3); yet, they are necessary for quantifying an advice-augmented online algorithm to be optimal. We now concretely explain the model, our algorithm, and the Pareto-efficient frontier it achieves between robustness and consistency.

36th Conference on Neural Information Processing Systems (NeurIPS 2022).

## 1.1 Model and Results

We study the two-stage version of the maximum vertex-weighted bipartite matching problem (see Subsection 1.2 for other variants). In this problem, there is an underlying bipartite graph $G = (D, S, E)$, where $D$ stands for "demand" and $S$ stands for "supply". The vertices in $S$ are weighted and *offline* (i.e. known in advance), while the vertices in $D$ arrive *online* in two batches $D_1$ and $D_2$. After the vertices in each batch arrive, all their incident edges to $S$ are revealed, and the algorithm chooses a matching in the resulting subgraph. (In particular, note that the first time the algorithm needs to make a decision is after the entire batch $D_1$ has arrived.) The algorithm's choices are irrevocable, and the goal of the problem is to maximize the total weight of the matched offline vertices after both batches have arrived. Observe that the optimal second-stage decision is clearly to select the max-weight matching subject to the matching already chosen in the first stage; therefore the complexity of the problem boils down to finding a good first-stage matching.

In the advice-augmented model we introduce, the algorithm receives a *suggested matching* $A$ to make after the edges incident to the first-stage vertices $D_1$ have been revealed. Our model allows for an arbitrary advice $A$, although typically $A$ would be the matching that, for a given (probabilistic) prediction of $D_2$ and its incident edges, maximizes the (expected) total weight matched after both stages have arrived. (There is a different line of work (see [2]) that assumes access to an oracle that gives perfect advice and studies the minimum number of queries needed to obtain a given guarantee.)

We define consistency based on this suggested matching, and explain later how it reduces to the standard notion of consistency based on prediction error (see Subsection 1.2). That is, we define an algorithm's *consistency* as how well it performs relative to the best matching that could have been obtained by *exactly* following the advice in the first stage. Meanwhile, we define an algorithm's *robustness* using the standard notion of *competitive ratio*, which is the algorithm's performance relative to the optimal maximum-weight matching in hindsight. We then say that an algorithm is $C$-consistent and $R$-robust, respectively, if its consistency is at least $C$ and robustness is at least $R$ for *all* possible graphs $G$ and suggested matchings $A$. Formally, this requires

$$\inf_{G,A} \frac{\mathsf{ALG}(G, A)}{\mathsf{ADVICE}(G, A)} \geq C \quad \text{and} \quad \inf_{G,A} \frac{\mathsf{ALG}(G, A)}{\mathsf{OPT}(G)} \geq R, \tag{1}$$

where following the earlier descriptions, $\mathsf{ALG}(G, A)$ is the algorithm's expected performance when given advice $A$ and the graph ends up being $G$, $\mathsf{ADVICE}(G, A)$ is the performance from following advice $A$ exactly on graph $G$, and $\mathsf{OPT}(G)$ is the maximum-weight matching in graph $G$. Note that since $\mathsf{ADVICE}(G, A) \leq \mathsf{OPT}(G)$, any algorithm that is $R$-robust is automatically at least $R$-consistent. However, $C$ can be higher than $R$, and having a consistency guarantee is desirable when the advice is good, because intuitively $\mathsf{ADVICE}(G, A)$ will be large and close to $\mathsf{OPT}(G)$.

We now describe our algorithm. Here, it is convenient to consider the *fractional relaxation* of the problem, which allows vertices to be filled partially. In Section 2 we formally define the fractional problem and show how to losslessly round any fractional solution to a randomized matching, so that we can focus on defining our algorithm for the fractional problem.

First, in the absence of advice (i.e. when one only cares about robustness), traditional algorithms for online matching can be described as using a *penalty function* to "balance" how much to fill the offline vertices; the optimal competitive ratio is usually achieved by carefully choosing the right penalty function. Using a linear penalty function, Feng et al. [1] give an algorithm for two-stage bipartite matching that is $\frac{3}{4}$-robust, which they prove is optimal. However, since this algorithm makes no use of the advice, it is only $\frac{3}{4}$-consistent. On the other hand, note that the naive algorithm which always follows the advice exactly is 1-consistent but 0-robust. At this point, a natural idea for obtaining a continuum of robustness-consistency guarantees is to run the algorithm of Feng et al. [1] with probability $p$, and otherwise run the naive algorithm with probability $1 - p$. As the coin-flip probability $p$ ranges from 0 to 1, the robustness/consistency of this coin-flip algorithm ranges along the line between $(R, C) = (0, 1)$ and $(R, C) = (\frac{3}{4}, \frac{3}{4})$ (see the orange dashed line in Figure 1).

To improve upon these naive guarantees, we revisit the penalty function approach. Our main insight is that for any given robustness level $R \in [0, \frac{3}{4}]$, there is actually a *family* of penalty functions that guarantees $R$-robustness. While a traditional algorithm for online matching would choose the same penalty function for all the offline vertices, we observe that by choosing *different* penalty functions, we can incentivize the algorithm to prioritize matching to the vertices that are suggested by the advice, all while guaranteeing $R$-robustness. We prove that such an algorithm ends up being

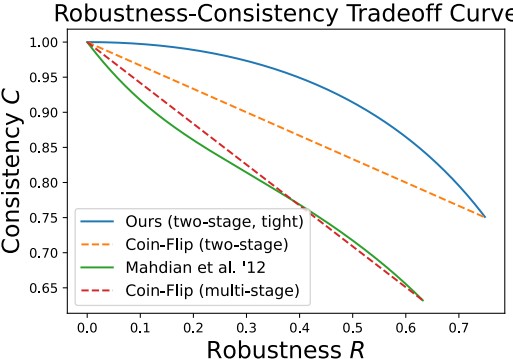

Figure 1: Robustness-consistency tradeoff curves for various algorithms.

$C$-consistent, where $C$ is the unique value (decreasing in $R$) that satisfies the equation $\sqrt{1-R} + \sqrt{1-C} = 1$. This curve is also plotted in Figure 1 and is a significant improvement over the straight-line interpolation. This elementary, symmetric curve also happens to be tight.

**Organization of paper.** Notation and the reduction from integer to fractional matching is presented in Section 2. Our algorithm is formally defined in Section 3. Proofs of robustness and consistency are sketched in Section 4 and Section 5 respectively, highlighting our new techniques. The hardness instance certifying our tradeoff curve of $\sqrt{1-R} + \sqrt{1-C} = 1$ to be tight is presented in Section 6. For the full, most up-to-date version of the paper which includes all proofs, please refer to https://arxiv.org/abs/2206.11397.

### 1.2 Discussion of Other Variants and Definitions

**Multi-stage variant.** Classical models of online matching and budgeted allocation [3, 4] allow more than two stages of online arrivals, and algorithms that *hybridize* between a fixed "advice" algorithm and traditional algorithms that "balance" against the adversary were introduced in Mahdian et al. [5]. However, under this framework it is difficult to define how the advice algorithm should react if the observed arrivals start to diverge from its predictions, making any tightness results between robustness and consistency elusive. Instead, we focus on the elegant two-stage model of Feng et al. [1], avoiding any complexities in how the advice should adapt to the accuracy of past predictions.

We note that the guarantees in Mahdian et al. [5] do not apply to our setting, since they do not handle the innovation in Feng et al. [1] of allowing multiple online vertices to arrive simultaneously. However, even supposing they could be generalized with no loss, their guarantees would be significantly worse than ours (see Figure 1, which plots in green the tradeoff curve described in Theorem 4.1 in their paper). In fact, as observed in the upper-left part of Figure 1, the guarantees of Mahdian et al. [5] can be worse than the simplistic coin-flip algorithm that either runs the "advice" algorithm, or runs the "balance" algorithm from the multi-stage setting[1]. This demonstrates the need for a simpler advice-augmented online matching framework as we propose, in which the Pareto-efficient tradeoff between robustness and consistency can be exactly understood.

**Unweighted variant.** In the special case where all offline vertices have weight 1, a better robustness-consistency tradeoff is possible—in fact, the simplistic coin-flip algorithm is optimal. This is because in the unweighted case, any maximal matching is at least $\frac{1}{2}$ as large as optimal, and therefore naively following the advice is now $\frac{1}{2}$-robust (instead of 0-robust) and still 1-consistent. The robustness-consistency guarantees are now defined by the straight line between $(R, C) = (\frac{1}{2}, 1)$ and $(R, C) = (\frac{3}{4}, \frac{3}{4})$, which is optimal.[2] However, in the vertex-weighted variant the straight-line guarantees are suboptimal, justifying the need for our approach in Subsection 1.1.

---

[1]In the multi-stage setting, the analogue of "balance" is only $(1 - \frac{1}{e})$-robust, hence the guarantees for the coin-flip algorithm now range along the line between $(R, C) = (0, 1)$ and $(R, C) = (1 - \frac{1}{e}, 1 - \frac{1}{e})$ (see the red dashed line in Figure 1).

[2]The instance which shows that this straight line tradeoff is tight for the unweighted setting is similar to the instance in Section 6, and can be found in the full version of the paper.

**Edge-weighted variant.** In this variant, the edges of the bipartite graph are weighted, and the aim is to maximize the total weight of the edges in the matching. For 2-stage edge-weighted bipartite matching, the naive coin-flip algorithm also turns out to be best-possible: The optimal tradeoff curve is the line segment between $(R, C) = (0, 1)$ and $(\frac{1}{2}, \frac{1}{2})$. To see this, first note that the best possible robustness is $\frac{1}{2}$. This is achieved by an algorithm that 1) with probability $\frac{1}{2}$, finds the maximum matching $M_1$ in the first stage and does nothing in the second stage, and 2) with probability $\frac{1}{2}$, does nothing in the first stage and finds the maximum matching $M_2$ in the second stage.[3] Thus, the coin-flip algorithm which naively interpolates between the $\frac{1}{2}$-robust algorithm and the 1-consistent algorithm (that always follows the advice) attains the tradeoff.

To show the tradeoff is tight, consider a first-stage graph consisting of a single edge with weight 1, and suppose the advice suggests matching the edge. Let $x$ be the probability the algorithm matches the edge. Any $R$-robust algorithm must have $x \leq 1 - R$, because if $x > 1 - R$ then we cannot maintain $R$-robustness if in the second stage, a single edge to the offline vertex arrives with very high weight. Since $x \leq 1 - R$, this means the maximum consistency on this instance is $1 - R$ (which is the case if the second-stage graph is empty). Thus, just as in the unweighted case, a straight-line tradeoff is also optimal in the edge-weighted variant.

**Consistency defined based on prediction error.** Many papers on prediction-augmented online algorithms (reviewed in Subsection 1.3) do not analyze two separate ratios $\frac{\mathsf{ALG}(G,A)}{\mathsf{OPT}(G)}$ and $\frac{\mathsf{ALG}(G,A)}{\mathsf{ADVICE}(G,A)}$. Instead, they provide a general guarantee on $\frac{\mathsf{ALG}(G,A)}{\mathsf{OPT}(G)}$ parameterized by *prediction error*, and define consistency as the guarantee obtained when the error is 0. The prediction error captures how much the prediction deviates from the truth, and its precise definition depends on the problem at hand.

Our definition of robustness and consistency can be easily translated to a guarantee involving prediction error, because we can always define the error of advice $A$ to be $\eta := 1 - \frac{\mathsf{ADVICE}(G,A)}{\mathsf{OPT}(G,A)}$. When we do so, our robustness/consistency guarantee immediately implies the parameterized guarantee $\frac{\mathsf{ALG}(G,A)}{\mathsf{OPT}(G)} \geq \max\{R, (1-\eta)(2\sqrt{1-R} - (1-R))\}$, for any value of $R \in [0, \frac{3}{4}]$ that we can set. More generally, our definition can accommodate other notions of predictions and their associated error, as long as it is possible to quantify how they affect the value of $\mathsf{ADVICE}(G, A)$.[4]

## 1.3 Further Related Work

Despite the recent surge of interest in both online matching and online algorithms with advice (see the recent surveys Huang and Tröbst [7] and Mitzenmacher and Vassilvitskii [8] respectively), literature on online matching with advice has been relatively scant. We mention some papers in this intersection below, as well as other related work.

**Online matching with advice.** Since Mahdian et al. [5], more recently Antoniadis et al. [9] have introduced an online random-order edge-weighted bipartite matching problem with advice that predicts the edge weights adjacent to each offline vertex in some optimal offline matching. Lavastida et al. [6] introduced a framework that formalizes when predictions can be learned from past data, and their algorithmic performance degrades gracefully as a function of prediction error. Broadly speaking, our work contrasts these three papers in that we are able to understand *tight* tradeoffs in whether to trust some advice, although in an arguably simpler setting.

**Online matching models in-between adversarial and stochastic.** Instead of abstracting all information about the future into a single piece of prediction/advice that could be wrong, another approach is to postulate an explicit model for how the future can deviate from past observations. Examples of this include the semi-online bipartite matching model of Kumar et al. [10] and the partially predictable model of Hwang et al. [11]. We note that random-order arrivals can also be viewed as a form of partial predictability which allows learning [12], and moreover it is possible to derive simultaneous guarantees under adversarial and random-order arrivals [13] which have the

---

[3]This algorithm is $\frac{1}{2}$-robust because the value of the optimal matching is at most the sum of the values of $M_1$ and $M_2$. On the other hand, it is an easy exercise to find an example which shows that no algorithm can be more than $\frac{1}{2}$-robust.

[4]As an example, Lavastida et al. [6] measure prediction error using a notion of total variation distance on the number of each "type" of online vertex to arrive. They propose algorithms that take advice in the form of a proportional allocation weight for each offline vertex.

same flavor as robustness-consistency guarantees. Finally, we mention that the single sample model in Kaplan et al. [14] can be viewed as a form of online matching with advice that is highly erroneous.

**Optimality results in prediction-augmented online algorithms.** Tight robustness-consistency tradeoffs have become recently understood in prediction-augmented ski rental [15–17] and single-commodity accept/reject problems [18, 19]. Our online matching problem contrasts these by having a *multi-dimensional* state space, for which to our knowledge tightness results are rare. We should mention that in multi-dimensional problems such as prediction-augmented caching [20] and online welfare maximization [21], "optimality" results in which consistency can be achieved with no loss of robustness (i.e. there is no "tradeoff") have been derived.

**Two-stage models.** Two-stage models capture the essence of optimization under uncertainty, where the first-stage decision must anticipate the uncertainty, and the second-stage decision is usually a trivial recourse after the uncertainty (in our case the second-stage graph) has been realized. We refer to Birge and Louveaux [22] and Bertsimas et al. [23] for broad overviews of two-stage stochastic and robust optimization. In this paper we focus on the two-stage online matching model of Feng et al. [1], introducing advice to this model and fully characterizing the tradeoff between obeying vs. disobeying the advice. We note that a multi-stage online matching model motivated by batching has also been recently considered in Feng and Niazadeh [24], and two-stage matching has been formulated as a robust optimization problem in Housni et al. [25].

## 2   Preliminaries and Notation

The problem input consists of a bipartite graph $G = (D, S, E)$, where the online vertices in $D$ arrive in two batches $D_1, D_2$. We index the vertices in $D$ (resp. $S$) with $i$ (resp. $j$). When the vertices in $D_k$ ($k = 1, 2$) arrive, all their incident edges to $S$ are revealed, and the algorithm irrevocably chooses a matching $M_k$ between $D_k$ and $S$. Each offline vertex $j \in S$ has a weight $w_j$, and the goal is to maximize $\sum_{(i,j) \in M_1 \cup M_2} w_j$. We evaluate an algorithm by its robustness and consistency, which were defined in Subsection 1.1.

**Worst-case instances.** The following observation is crucial to our analysis. Note that when bounding robustness and consistency, one can assume WLOG that the second-stage graph consists of a matching. To see this, suppose we are bounding robustness, and consider the edges *not* matched by $\mathsf{OPT}(G)$ in the second stage. Deleting these edges does not change the value of $\mathsf{OPT}(G)$ and can only decrease the value of the matching found by the algorithm. Therefore we may assume that the second stage graph consists *exactly* of the matching selected by $\mathsf{OPT}(G)$ in the second stage. The same argument shows that when bounding consistency, we may assume that the second-stage graph consists exactly of the matching selected by $\mathsf{ADVICE}(G, A)$ in the second stage. Therefore from now on we assume the second-stage graph is a matching.

**Reduction to fractional matching.** The preceding observation about worst-case instances allows us to focus on the fractional version of the problem, which is easier to analyze and defined as follows. In each stage, the algorithm chooses a *fractional matching* instead of an integral one. Let $\mathbf{x} = (x_{ij} : i \in D_1, j \in S, (i,j) \in E)$ be the fractional matching chosen in the first stage, so that $\mathbf{x}$ satisfies the constraints $x_i := \sum_{j:(i,j) \in E} x_{ij} \leq 1$ and $x_j := \sum_{i:(i,j) \in E} x_{ij} \leq 1$. Similarly, let $\mathbf{y} = (y_{ij} : i \in D_2, j \in S, (i,j) \in E)$ be the fractional matching in the second stage, which satisfies $y_i := \sum_{j:(i,j) \in E} y_{ij} \leq 1$ and $y_j := \sum_{i:(i,j) \in E} y_{ij} \leq 1 - x_j$. The objective in the fractional problem is to maximize $\sum_{j \in S} w_j(x_j + y_j)$. As mentioned earlier, the first stage decisions $x_j$ are the critical ones; therefore we will use the terminology that each offline vertex $j \in S$ is *filled to water level $x_j$* at the end of the first stage.

Although the fractional problem seems at first to be easier than the integral problem, it turns out to be straightforward to convert any fractional algorithm for two-stage bipartite matching into a (randomized) integral one with the same robustness/consistency guarantees. To see why, consider any fractional algorithm and let $\mathbf{x}$ be its first-stage output. To convert this to a randomized integral algorithm, we sample an integral matching $M_1$ with marginals equal to $\mathbf{x}$ (i.e. $\mathbb{P}((i,j) \in M) = x_{ij}$

---
**ALGORITHM 1:** Algorithm for Two-Stage Fractional Vertex-Weighted Bipartite Matching with Advice
---
**Input:** Suggested matching $A$ in the first-stage graph and desired robustness level $R$.

1: (Define penalty functions) Let $f_{\mathsf{L}}(x) = \max\{0, 1 - \frac{1-R}{x}\}$ and $f_{\mathsf{U}}(x) = \min\{1, \frac{1-R}{1-x}\}$.

2: (First stage) When the first-stage vertices $D_1$ arrive, let $S_1 \subseteq S$ be the set of offline vertices that are in the suggested matching $A$. Set $f_j = f_{\mathsf{L}}$ for all $j \in S_1$ and $f_j = f_{\mathsf{U}}$ for all $j \in S \setminus S_1$.
Solve the following optimization problem for the first-stage fractional matching $\bar{\mathbf{x}}$:

$$(P_1) \quad \max \sum_{j \in S} w_j \left( x_j - \int_0^{x_j} f_j(t)dt \right)$$

$$\text{s.t. } x_i := \sum_{j:(i,j)\in E} x_{ij} \leq 1 \qquad\qquad \forall\, i \in D_1$$

$$x_j := \sum_{i \in D_1:(i,j)\in E} x_{ij} \leq 1 \qquad\qquad \forall\, j \in S$$

$$x_{ij} \geq 0 \qquad\qquad \forall\, i \in D_1, (i,j) \in E$$

3: (Second stage) When the second-stage vertices $D_2$ arrive, solve for the optimal fractional matching $\bar{\mathbf{y}}$ subject to the capacities already taken by $\bar{\mathbf{x}}$:

$$(P_2) \quad \max \sum_{j \in S} w_j y_j$$

$$\text{s.t. } y_i := \sum_{j:(i,j)\in E} y_{ij} \leq 1 \qquad\qquad \forall\, i \in D_2$$

$$y_j := \sum_{i \in D_2:(i,j)\in E} y_{ij} \leq 1 - \bar{x}_j \qquad\qquad \forall\, j \in S$$

$$y_{ij} \geq 0 \qquad\qquad \forall\, i \in D_2, (i,j) \in E$$

4: Return $\bar{\mathbf{x}} + \bar{\mathbf{y}}$.
---

for all edges $(i, j)$ in the first-stage graph).[5] We then take $M_2$ to be the maximum-weight matching in the second-stage graph, subject to $M_1$ already being chosen.

To analyze the (expected) weight of the integral matching $M_1 \cup M_2$, note that in the first stage the expected weight of $M_1$ is equal to that of $\mathbf{x}$ by construction. Now consider the second stage. As we have argued, we may assume the second-stage graph consists of a matching; let $(i, j)$ be one of these edges. The contribution of $(i, j)$ to the value of the fractional algorithm is $w_j(1 - x_j)$, because the remaining amount that offline vertex $j$ can be filled is $(1 - x_j)$. On the other hand, the integral algorithm will match $(i, j)$ in $M_2$ if and only if $j$ is unmatched in $M_1$, which happens with probability $(1 - x_j)$. So the contribution of $(i, j)$ to the expected increase of the integral algorithm is also $w_j(1 - x_j)$. Since the integral version of the problem can be reduced to the fractional version, we focus on the latter in the remainder of the paper.

## 3 Algorithm

As argued in Section 2, we can losslessly round a fractional matching to an integral one. Therefore we focus on the *fractional* two-stage bipartite matching problem from now on. Our algorithm is described in Algorithm 1. The main challenge is to commit to a matching $\mathbf{x}$ when the first-stage graph is revealed, that incorporates some advice but maintains a worst-case competitive ratio.

Intuitively, the amount $x_j$ that each vertex $j \in S$ is filled to after the first stage is governed by three factors. First, it should depend on the weight $w_j$; the larger $w_j$ is, the larger $x_j$ should be. Second, it should depend on the advice; $x_j$ should be larger if $j$ is recommended by the advice, and smaller if not. Third, it should depend on how much the *other* offline vertices are filled; the larger $x_j$ is compared to the other vertices competing to be filled, the more hesitant we should be of filling it even further, because this can be exploited by a second-stage graph whose edges are incident to the

---

[5]Since the bipartite matching polytope is integral, $\mathbf{x}$ can be written as a convex combination of integral matchings. Such a convex combination can be found in polynomial time using algorithmic versions of Carathéodory's theorem.

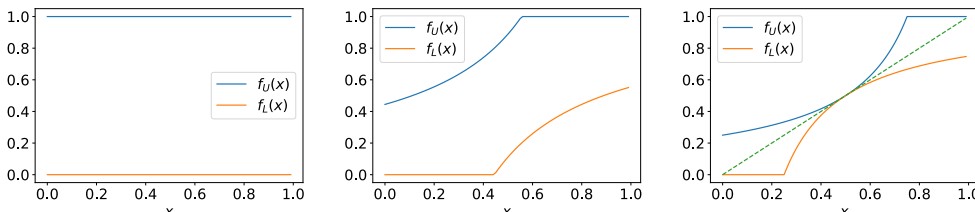

Figure 2: Plots of $f_L$ and $f_U$ for three values of $R$. Left: $R = 0$. Middle: $R = \frac{5}{9}$. Right: $R = \frac{3}{4}$. The consistencies achieved are $C = 1, C = \frac{8}{9}, C = \frac{3}{4}$, respectively. In the right plot, the green dashed line indicates the penalty function used by Feng et al. [1].

vertices which have already been filled the most. Typically, the third point is achieved by defining a *penalty* $f(x_j) \in [0, 1]$ on the marginal benefit of filling $j$, with $f(x_j)$ increasing in $x_j$, which encourages the balancing of the water levels across $j \in S$. This is combined with the first point by associating a *potential* $w_j(1 - f(x_j))$ with each $j \in S$. The algorithm then decreases these potentials while filling water levels, with the aim of equalizing them as much as possible.

Our main idea to address the second point, now that there is some advice, is to use a *lower* penalty rate $f_L(x_j)$ to incentivize the filling of offline vertices $j \in S_1$ that are matched by the advice, and a *higher* penalty rate $f_U(x_j)$ for $j \notin S_1$. More specifically, for any desired "robustness" $R$, our analysis characterizes the envelope of penalty functions $f : [0, 1] \to [0, 1]$ (see Theorem 1) that guarantees a worst-case competitive ratio of $R$, and defines $f_L, f_U$ according to the lower, upper boundaries of this envelope respectively.

Figure 2 plots the penalty functions $f_L$ and $f_U$ for three values of $R$. The parameter $R \in [0, \frac{3}{4}]$ affects the separation between $f_L$ and $f_U$. If $R = 0$, then $f_L, f_U$ are the constant functions $0, 1$ respectively, from which it can be seen that Algorithm 1 will fully match all the vertices suggested by the advice. Intuitively, setting $R = 0$ is desirable when we trust that the advice will perform well. On the other extreme, setting $R = \frac{3}{4}$ guarantees the maximum possible robustness, which is not reliant on the advice. Surprisingly, even in this case there is a bit of separation between $f_L$ and $f_U$ (see Figure 2), and hence the algorithm still discriminates between offline vertices $j \in S_1$ vs. $j \notin S_1$, despite its distrust in the advice. We plot one intermediate value where $R = \frac{5}{9}$.

Finally, having decided penalty functions for the offline vertices, there is still the question of how to balance their potential levels when multiple online vertices $D_1$ can arrive in the first stage. Fortunately, this question was answered by Feng et al. [1], who show that problem $P_1$ is the right one to solve, and establish a neat structural decomposition on its solution. Our Lemma 1 is a simplified version of the structural decomposition from their paper, which suffices for our purposes, and importantly, allowing for *heterogeneous* penalty functions across offline vertices $j \in S$.

**Definition 1.** *Let $\bar{\mathbf{x}}$ be the first-stage fractional matching found by Algorithm 1. For all $j \in S$, we say $j$ is **filled** if $\bar{x}_j = 1$ or $f_j(\bar{x}_j) = 1$. Let $F$ denote the set of filled offline vertices, and let $U = S \setminus F$ denote the set of **unfilled** offline vertices.*

**Lemma 1.** *After the first-stage fractional matching, the following statements hold for all $i \in D_1$:*

1. *If $i$ has any unfilled neighbor (i.e. $j \in U$ for some $j$ with $(i, j) \in E$), then $\bar{x}_i = 1$.*

2. *For any unfilled neighbor $j$ and any neighbor $k$ with $\bar{x}_{ik} > 0$, it must be that*

$$w_j(1 - f_j(\bar{x}_j)) \le w_k(1 - f_k(\bar{x}_k)). \tag{2}$$

We explain the intuition behind Lemma 1. Property 1 holds because if $\bar{x}_i < 1$ instead, then variable $\bar{x}_{ij}$ where $j$ is any unfilled neighbor of $i$ could have been increased to strictly improve the objective of $(P_1)$. Meanwhile, Property 2 holds because if $\bar{x}_{ik} > 0$, i.e. $i$ is being used to fill $k$ in the current solution, then the potential of $k$ must be no less than the potential of any unfilled neighbor $j$ that could have been filled instead. We note that the structural decomposition in Feng et al. [1] is richer, but our simplified properties suffice for our purposes and are easier to establish when our penalty functions $f_j$ (due to heterogeneity) do not necessarily satisfy their boundary conditions such as $f_j(0) = 0$ or $f_j(1) = 1$. For space reasons, the proof of Lemma 1 is in the full version of the paper.

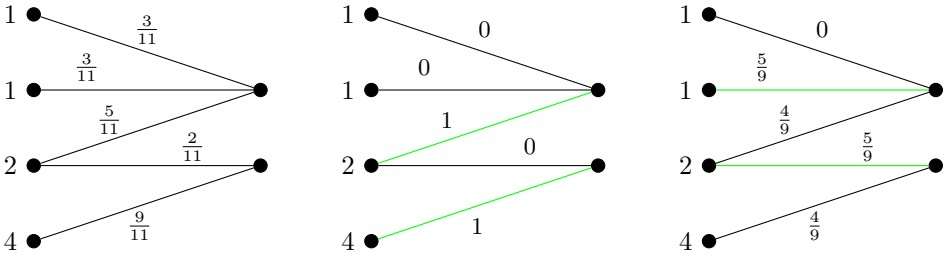

(a) The fully robust solution.  (b) The algorithm's first-stage matching if the advice suggests $(1,3),(2,4)$.  (c) The algorithm's first-stage matching if the advice suggests $(1,2),(2,3)$.

Figure 3: Example to illustrate the features of the algorithm. $S$ is on the left and $D_1$ is on the right. The vertices in $S$ are $i = 1, 2, 3, 4$ and the vertices in $D_1$ are $j = 1, 2$ (labelling goes from top to bottom). The number next to $j \in S$ is its weight $w_j$. The green edges are suggested by the advice.

## 3.1 Illustration of how our Algorithm uses Advice

As an example, suppose $R = \frac{5}{9}$ and $C = \frac{8}{9}$, where we note that $\sqrt{1-R} + \sqrt{1-C} = 1$. Let $D_1$ consist of vertices $i = 1, 2$ and $S$ consist of vertices $j = 1, 2, 3, 4$ with weights $w_j = 1, 1, 2, 4$ respectively. The edges in the graph are $(1,1), (1,2), (1,3)$ and $(2,3), (2,4)$.

We first demonstrate the case where the advice suggests edges $(1,3)$ and $(2,4)$, which would match weight $w_3 + w_4 = 6$ in the first stage. Intuitively, this advice is not so "extreme" in that it is greedily matching the highest-weight vertices in $S$ while it can, without assuming it can later match them in the second stage. In this case our algorithm would follow the advice exactly (see Figure 3b), and have a consistency of 1. Its robustness would be at least $\frac{6}{7}$, with the worst case being when the second stage consists of a single edge $(3,4)$, which cannot be matched by our algorithm but increases the optimal offline matching from 6 to 7. Nonetheless, this suffices because $\frac{6}{7}$ is well above the targeted robustness of $R = \frac{5}{9}$.

We now demonstrate the case where the advice suggests edges $(1,2)$ and $(2,3)$ instead, perhaps predicting that an edge $(3,4)$ will allow us to match offline vertex 4 later. This advice is quite "extreme" in that it is skipping the highest-weight vertex in $S$ in the first stage, based on a second-stage prediction which may not come to fruition. Following it exactly would give a weight of $w_2 + w_3 = 3$ in the first stage, which cannot be more than $\frac{3}{7}$-robust[6], significantly lower than the target of $\frac{5}{9}$. Meanwhile, the maximally robust fractional matching based on linear penalty functions (Feng et al. [1]; shown in Figure 3a), which judiciously balances between *all* the offline vertices, is not $\frac{8}{9}$-consistent[7]. Our algorithm returns the solution in Figure 3c, which follows the advice in that it completely prioritizes vertex 2 over vertex 1, but deviates by significantly filling offline vertex 4 (which has the highest weight) as a failsafe (although not as much as the fully robust solution). This makes it both $\frac{5}{9}$-robust and $\frac{8}{9}$-consistent, and our algorithm can be adjusted accordingly to be both $R$-robust and $C$-consistent for any values $R, C$ satisfying $\sqrt{1-R} + \sqrt{1-C} \leq 1$.

Based on these examples, we highlight two desirable features of our algorithm. First, it naturally responds to the "extremity" of the advice, by deviating more from the more extreme advice (in the 2nd case) in order to maintain $R$-robustness. Second, from the definition of $f_U$ and $f_L$ in Algorithm 1 together with Lemma 1, it can be seen that if $R \leq \frac{1}{2}$ and the graph is unweighted, then our algorithm will *always* follow the advice exactly. This is because $f_L(1) = R \leq 1 - R = f_U(0)$, which implies the algorithm will prioritize filling a vertex suggested by the advice over one that is not, even if the former is completely filled and the latter is empty. Put another way, the algorithm automatically recognizes that any maximal matching will be at least $\frac{1}{2}$-robust in an unweighted graph.

---

[6] The worst case is if the second-stage graph consists of a single edge $(3,2)$, in which case $\mathsf{ALG}(G,A) = 3$ and $\mathsf{OPT}(G) = 7$.

[7] If the second-stage graph consists of edges $(3,1)$ and $(4,4)$, then the fully robust solution gets a value of $1 + \frac{3}{11} + 2 \cdot \frac{7}{11} + 4 = \frac{72}{11}$ whereas $\mathsf{ADVICE}(G,A) = 8$. Their ratio is $\frac{9}{11}$, which is less than $\frac{8}{9}$.

# 4 Analysis of Robustness

We begin by analyzing the robustness of Algorithm 1, which is its competitive ratio with respect to the optimal matching in hindsight. The theorem below gives a characterization of the penalty functions $f_j$ that are sufficient to guarantee $R$-robustness, which is what led to our plots in Figure 2.

**Theorem 1.** *Let $R \in [0, \frac{3}{4}]$. Suppose we run Algorithm 1 where the functions $f_j$ satisfy the following properties:*

1. *$f_j : [0, 1] \to [0, 1]$ is increasing,*

2. *$1 - \frac{1-R}{x} \leq f_j(x) \leq \frac{1-R}{1-x}$.*

*Then Algorithm 1 is $R$-robust.*

The proof of Theorem 1 employs the online primal-dual technique. Based on the algorithm's decisions, we construct dual variables that are approximately feasible and lower bound the algorithm's objective value. (This is also the proof approach used by [1], but we note that the way our dual variables are set is necessarily different in order to derive our results. ) The simplified structural property (Lemma 1) allows us to characterize the envelope of functions that guarantee $R$-robustness in the primal-dual analysis. Roughly speaking, the upper bound on $f_j$ guarantees approximate dual feasibility for the first stage edges, while the lower bound on $f_j$ guarantees it for the second stage edges. For space reasons, we have deferred the proof to the full version of the paper.

# 5 Analysis of Consistency

We now turn to analyzing the consistency of the algorithm. In the previous section, we described an envelope of functions (see Figure 2) that were sufficient to guarantee $R$-robustness. For a desired robustness level $R$, we now maximize consistency by setting the penalty functions $f_j$ to their lower bound for vertices suggested by the advice, and to their upper bound for vertices not suggested by the advice. The theorem below shows that this choice of the penalty functions achieves the desired robustness-consistency tradeoff. Later in Section 6 we show this tradeoff to be tight.

**Theorem 2.** *Let $R \in [0, \frac{3}{4}]$, and let $S_1$ be the set of offline vertices that are matched by the suggested matching in the first-stage graph. Suppose we run Algorithm 1 with*

- *$f_j(x) = \min\{\frac{1-R}{1-x}, 1\}$ for all $j \in S \setminus S_1$, and*

- *$f_j(x) = \max\{1 - \frac{1-R}{x}, 0\}$ for all $j \in S_1$.*

*Then Algorithm 1 is $(2\sqrt{1-R} - (1-R))$-consistent.*

Our high-level strategy for bounding consistency (which recall is $\frac{\mathsf{ALG}(G,A)}{\mathsf{ADVICE}(G,A)}$) is to carefully split the value of $\mathsf{ALG}(G, A)$, attributing a separate contribution to each first-stage vertex $i \in D_1$. We then split the value of $\mathsf{ADVICE}(G, A)$ in a similar way. This decomposes the numerator and denominator as sums over $i \in D_1$, which means it suffices to lower-bound the ratio of the terms corresponding to each individual $i \in D_1$. For concreteness, consider some fixed $i \in D_1$ and let $a \in S$ be the offline vertex that the advice suggests matching $i$ to. The bad case for consistency is if $\bar{x}_a$ is low, because this means the algorithm deviated from the advice. However, because the potential functions are chosen to *incentivize* matching to $a$, the only way $\bar{x}_a$ can be low is if $i$ is substantially matched to other vertices with higher weight than $w_a$, implying the algorithm must have collected a lot of value from the vertices that were not recommended by the advice. The full proof, which makes this intuition precise, is deferred to the full version of the paper for space reasons.

# 6 Tightness of the Robustness-Consistency Tradeoff

Theorems 1 and 2 together show that Algorithm 1 is able to be both $R$-robust and $C$-consistent, for any $(R, C)$ with $R \in [0, \frac{3}{4}]$ and $\sqrt{1-R} + \sqrt{1-C} = 1$. We now show this tradeoff to be tight.

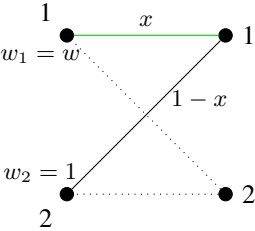

Figure 4: Illustration of the hardness instance. $S$ is on the left and $D$ is on the right. The first arrival neighbors both vertices of $S$. The second arrival neighbors exactly one vertex of $S$, but it could be either vertex. The advice is to match the green edge $(1, 1)$.

**The hardness instance.** The hardness instance is illustrated in Figure 4. It is a graph where $D$ has two vertices $i = 1, 2$ that arrive one in each stage. $S$ has two vertices $j = 1, 2$ with weights $w_1 = w$ (for some $w \in [0, 1]$ that we will set later) and $w_2 = 1$. The first-stage graph consists of both edges $(1, 1)$ and $(1, 2)$, and the advice suggests matching along the edge $(1, 1)$, which is "extreme" advice in that offline vertex 1 has the lower weight one edge (which can be either $(2, 1)$ or $(2, 2)$). Since the algorithm does not know which edge will arrive in the second stage, it must hedge against both possibilities when making its first-stage decision.

**Theorem 3.** *Let $w = \frac{1}{\sqrt{1-R}} - 1$ in the instance described above.[8] Then any algorithm that is $R$-robust is at most $C$-consistent where $C = 2\sqrt{1 - R} - (1 - R)$.*

*Proof of Theorem 3.* Let $x := x_{11}$ and $1 - x := x_{12}$, so that the algorithm's first-stage decision is entirely characterized by the value of $x$. There are two cases.

1. Edge $(2, 1)$ arrives in the second stage. Then $\mathsf{ALG} = w + 1 - x$ and $\mathsf{ADVICE} = w$.

2. Edge $(2, 2)$ arrives in the second stage. Then $\mathsf{ALG} = wx + 1$ and $\mathsf{ADVICE} = 1 + w$.

(Note that regardless of which edge arrives in the second stage, $\mathsf{OPT}$ is always equal to $1 + w$.) For the algorithm to be $R$-robust in Case 1, we must have

$$\frac{w + 1 - x}{1 + w} \geq R \implies x \leq (1 - R)(1 + w).$$

On the other hand, for the algorithm to be $C$-consistent in Case 2, we must have

$$\frac{wx + 1}{1 + w} \geq C \implies x \geq \frac{C(1 + w) - 1}{w}.$$

Since the algorithm does not know which of the two cases will happen in the second stage, it must choose an $x$ that satisfies both of the inequalities above. For a desired robustness $R$ and consistency $C$, this is only possible if

$$\frac{C(1 + w) - 1}{w} \leq (1 - R)(1 + w),$$

which when rearranged becomes

$$C \leq (1 - R)w + \frac{1}{1 + w}.$$

Substituting $w = \frac{1}{\sqrt{1-R}} - 1$ above gives $C \leq 2\sqrt{1 - R} - (1 - R)$, as desired. $\square$

---

[8]This choice of $w$ was obtained by minimizing $(1 - R)w + \frac{1}{1+w}$ over $w$.

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
