# OpenReview forum: "Online Bipartite Matching with Advice: Tight Robustness-Consistency Tradeoffs for the Two-Stage Model"
_NeurIPS.cc/2022/Conference — NeurIPS 2022 Accept_

### Official Review · Reviewer_iKQU · 2022-06-28

**Rating:** 6
**Confidence:** 4
**Soundness:** 4 excellent
**Presentation:** 2 fair
**Contribution:** 2 fair

**Summary:**

The paper studies the problem of augmenting the recently introduced two stage vertex-weighted online bipartite matching problem with predictions. In this problem one "sees" one side U of the bipartition. The vertices on the other side W are associated with a weight each and arrive in two batches. The algorithm has to decide on which edges to add to the matching after the first batch arrives and then do the same again after the second batch arrives (while still being consistent with previously selected edges). Goal is to maximise the total weight of vertices from W that are matched.

The authors argue that it is without loss of generality to assume that the vertices of the second batch only come with the edges that the optimal solution uses for these vertices, so the problem essentially reduces to handling the first batch. It is assumed that the algorithm has access to a prediction on how to match the first batch of vertices. This prediction may or may not be accurate, so the algorithm needs to keep a sort of tradeoff, between safe-guarding itself for all possible second batches that could still arrive (robustness) and following the predicted matching (in order to attain consistency, i.e., good performance when the prediction is accurate).

There is already a recent result by Feng et al. (SODA 2021) that looks at the problem without predictions. The proposed algorithm here identifies, for each desired robustness how to fine tune the parameters in Feng et al.'s  algorithm so as to obtain the best possible consistency that the desired robustness allows. A very simple example shows that the obtained robustness and consistency are Pareto optimal.

**Questions:**

Can you obtain similar results when the number of batches increases to 3? What about the "purely online" setting with "n batches"? I would find this a much more interesting problem/result. If not, where is the challenge?

**Limitations:**

No potential negative social impact. With respect to limitations, see above.

**Strengths And Weaknesses:**

Strengths:

- The matching problem considered is quite interesting.
- Although the algorithm is a slight modification of the one by Feng et al., there is still a non-negligible amount of technical work required to "set the parameters" correctly and prove the obtained results

Weaknesses:

- I found it quite hard to follow the paper. The main reasons I could identify for this are the following:
* The problem is not clear at all. The first time the paper mentions that they actually look into the *maximization* version is on page 5. I was until this point under the impression that a minimisation problem is considered which made many statements confusing (I assume if I were familiar with Feng's result it may have been clear, but in my opinion the authors should not assume familiarity with that result).
* The performance of the algorithm (also the one without predictions) must surely depend on the sizes of the respective batches D_1 and D_2, but this is not touched upon at all. It seems to me that this is handled by the way that consistency is defined (i.e., comparing to an algorithm that uses the same batches). The authors suggest a way to define an error measure to get a result as a function to the prediction quality, similar to other results in the area. This definition seems to me quite adhoc, and not very reasonable. The only justification seems to be that "this works" to give the desired formula.

- I find that predicting the whole matching for D_1 at once can be a bit unnatural.

- I am not sure if I would call the problem online. It resembles some similarity to online algorithms in that there must be an irrevocable decision after the first batch without knowledge of the second batch, but thats all. Also it should be made more explicit that the whole batch arrives before the algorithm needs to decide anything.

- The paper refers to the prediction as advice. This can be confusing, since in the online algorithms literature the term "advice" has historically been used for predictions that are guaranteed to be optimal (and usually one studies the relationship between the amount of that advice required to obtain a specific performance guarantee). When the predictions are of unknown quality as is the case here, the term "predictions" seems to be much more established and I would suggest to change the paper accordingly.

- No experimental evaluation is contained.

---

> ### Author Response · Authors · 2022-08-01
> **Response to Reviewer iKQU**
>
> Thank you for the review and questions!
>
> ---
>
> **1. Regarding the confusion about max-weight matching:**
>
> Sorry for the confusion.  We do say that we are maximizing the total weight of the matching several times in Section 1.1, but agree that this was mostly done in passing.  We have added a sentence in the revision to emphasize that we are studying max-weight matching, not min-weight perfect matching or one of those variants.
>
> ---
>
> **2. Regarding the dependence of the algorithm performance on the size of the batches, and also the definition of prediction error:**
>
> The performance guarantees of our algorithm do not depend on the sizes of $D_1$ and $D_2$, because the definitions of robustness and consistency take an infimum over all instances. Put another way, our guarantees hold *irrespective* of the sizes of the graphs.  In general it is quite common for these types of guarantees to not depend on the input size; for example, in the classic paper by Karp, Vazirani and Vazirani on online bipartite matching (STOC 1990), they give an algorithm whose competitive ratio is $1 - 1/e$, a guarantee that does not depend on the size of the graph.
>
> And thanks for commenting on the measure of prediction error we mention briefly before Section 1.3.  We just want to emphasize that the important aspect of this measure is that prediction error is 0 *if and only if* the prediction is sufficient to make an optimal decision.  You are totally right that the specific formula is perhaps ad hoc, but we want to emphasize that this "important aspect" is not ad hoc -- it is the case for almost all definitions of prediction error in the cited literature.  All in all, this means that our Consistency guarantee implies that ALG/OPT $\geq$ C whenever the prediction was sufficient to make an optimal decision -- which is the standard type of Consistency guarantee given in the literature.
>
> ---
>
> **3. Regarding the following comment:**
> >*I find that predicting the whole matching for D_1 at once can be a bit unnatural.*
>
> Since the entire batch $D_1$ arrives before the algorithm needs to make a decision, the prediction should depend on the entirety of $D_1$. This is why we take it to be a suggested matching in the whole first-stage graph, instead of, e.g. suggesting edges one at a time.
>
> ---
>
> **4. Regarding the use of the term "online":**
>
> Thanks for the comment. We fully agree that it is better to refer to the problem as two-stage bipartite matching wherever possible. We will also clarify explicitly in the revision that the algorithm only needs to make a decision after the entire batch has arrived.
>
> We see the two-stage problem as the simplest version of "online matching", and since understanding Robustness-Consistency tradeoffs is quite challenging in online matching, we wanted to focus on this parsimonious problem first.  So in that sense, we are the first to derive a tight Robustness-Consistency tradeoff for *any* variant of online matching.  That being said, we will make sure not to mislead the reader into thinking we have solved the problem for the general problem beyond two stages.
>
> ---
>
> **5. Regarding the terms "prediction" vs "advice":**
>
> Indeed, there are papers in the online algorithms literature that assume access to a perfect oracle and study the amount of information that is needed to achieve a desired guarantee, and we agree that in that setting it is common to refer to this perfect oracle as "advice". On the other hand, our paper studies imperfect predictions and the tradeoffs between algorithm performance and prediction error, and in this literature it is quite common to refer to the imperfect predictions as "advice". For examples of this, see the papers *Single-Leg Revenue Management with Advice* by Balseiro, Kroer, and Kumar and *Chasing Convex Bodies and Functions with Black-Box Advice*, by  Christianson,  Handina, and Wierman. We hope you are okay with our choice of terminology, and we have added a sentence to the revision to acknowledge that other literatures use the term "advice" differently.
>
> ---
>
> **6. Regarding the challenges of generalizing to multiple batches:**
>
> We agree that generalizing to multi-stage matching would be quite interesting! There is a natural generalization of our algorithm to multiple-stage fractional matching, but we do not know how to generalize the analysis. For the algorithm, one can say that there will be a suggested matching recommended in each stage. Then, set the penalty functions in each stage to be lower for the recommended vertices, and higher for the non-recommended vertices. One can then solve a convex optimization problem for the fractional matching in each stage, just like before. The challenge in generalizing the analysis is how to define ADVICE(G, A), since with multiple stages, the advice should be able to adapt to the decisions of the algorithm in previous stages.

---

> > ### Comment · Reviewer_iKQU · 2022-08-08
> > **Thanks**
> >
> > Thanks for the detailed and satisfactory response. Although I still find the model of predicting the whole of D1 and having only two stages a bit unnatural and it is arguably a very restricted version of the "online" setting, I see your point and have updated my score from a 5 to a 6.

---

### Official Review · Reviewer_d2AC · 2022-07-09

**Rating:** 7
**Confidence:** 4
**Soundness:** 4 excellent
**Presentation:** 4 excellent
**Contribution:** 3 good

**Summary:**

In this paper, the authors consider the two-stage bipartite matching problem in a relatively new setting: there is an unreliable prediction in the first stage. The goal is to design the online algorithm which can balance the consistency (performance in the best case) and the robustness (performance in the worst case). Their algorithm can achieve the optimal Pareto-efficient frontier between consistency and robustness. The key idea is to choose different penalty functions for different offline vertices. They prove that their algorithm can achieve R-robust and C-consistent if the value R,C satisfy the inequality \sqrt{1-R}+\sqrt{1-C}<=1. They also give the tight example to show that such tradeoff is optimal.

**Questions:**

I have a question about the claim “we can assume the second-stage graph is a matching” in section 2, page 5. If we only consider the measurement robustness (or consistency), we can assume the second stage graph is a matching, which is exactly the matching chosen by OPT (or ADVICE). However, if we consider both the robustness and the consistency, I am not convinced that we can assume the second stage graph is a matching. In the worst-case instance, it is possible that OPT and ADVICE chooses different matchings in the second stage graph. If this is the case, suppose we delete all edges which are not selected by OPT in the second stage graph, OPT(G) keeps the same while ALG(G,A) may decrease. However, in this case, we do not know the change of ADVISE(G,A), thus consistency C might be increase. I still think the claim should be correct, but the argument is not clean.

**Limitations:**

Yes. The authors clearly address the limitations. I don’t think there are potential negative societal impact of their work.

**Strengths And Weaknesses:**

Strengths:

1.	The result shows the clean and optimal tradeoff relation between consistency and robustness. This is a very nice theoretical result.
2.	It is nice to model the consistency as ALG/ADVISE. In many other papers, we will model consistency as the performance of algorithm when the prediction is accurate. It is difficult to argue tradeoff between consistency and robustness. Thus, the model in this paper is in some sense more equal between consistency and robustness.

Weaknesses:

1.	The results hold only for a simple setting of online matching, that is, they consider only vertex-weighted version and two-stage model. However, although it is unclear how to generalize the results to edge-weighted or multi-stage, it is an interesting start.

---

> ### Author Response · Authors · 2022-08-01
> **Response to Reviewer d2AC**
>
> Thank you for the review and questions!
>
> ---
>
> **Question 1:**
> *I have a question about the claim “we can assume the second-stage graph is a matching” in section 2, page 5. If we only consider the measurement robustness (or consistency), we can assume the second stage graph is a matching, which is exactly the matching chosen by OPT (or ADVICE). However, if we consider both the robustness and the consistency, I am not convinced that we can assume the second stage graph is a matching. In the worst-case instance, it is possible that OPT and ADVICE chooses different matchings in the second stage graph. If this is the case, suppose we delete all edges which are not selected by OPT in the second stage graph, OPT(G) keeps the same while ALG(G,A) may decrease. However, in this case, we do not know the change of ADVISE(G,A), thus consistency C might be increase. I still think the claim should be correct, but the argument is not clean.*
>
>
> **Response:** Thank you for this detailed question. The reason why it is not a problem that OPT and ADVICE choose different matchings in the second stage is because robustness and consistency are separate metrics. Our argument shows that 1) on every possible instance, our algorithm is at least R-robust and 2) on every possible instance, our algorithm is at least C-consistent.  When arguing about Robustness, we only have to consider the worst-case second stage for the metric of Robustness; meanwhile, when arguing about Consistency, we only have to consider the worst-case second stage for the metric of Consistency.  You are correct that these worst-case instances could be different, but our guarantees hold in spite of this.
>
> ---
>
> **Question 2:**
> *However, although it is unclear how to generalize the results to edge-weighted or multi-stage, it is an interesting start.*
>
> **Response:** We agree that generalizing to multi-stage matching would be quite interesting! There is a natural generalization of our algorithm to multiple-stage fractional matching, but we do not know how to generalize the analysis. For the algorithm, let there be a suggested matching recommended in each stage. Then, set the penalty functions in each stage to be lower for the recommended vertices, and higher for the non-recommended vertices. One can then solve a convex optimization problem for the fractional matching in each stage, just like before. The challenge in generalizing the analysis is how to define ADVICE(G, A), since with multiple stages, the advice should be able to adapt to the decisions of the algorithm in previous stages.
>
> Regarding generalizing to edge-weighted graphs, for the special case of 2-stage edge-weighted bipartite matching, we actually do know the tight tradeoff. The best possible tradeoff curve turns out to be the straight line between $(R, C) = (0, 1)$ and $(1/2, 1/2)$. In case you are interested, here is proof sketch:
>
> 1. First of all, note that the best possible robustness is $1/2$. This is achieved by an algorithm that 1) with probability $1/2$, finds the maximum matching in the first stage and does nothing in the second stage, and 2) with probability $1/2$, does nothing in the first stage and finds the maximum matching in the second stage. It is an easy exercise to find an example which shows that no algorithm can be more than $\frac12$-robust.
> 2. Taking a convex combination of the $\frac12$-robust algorithm with the 1-consistent algorithm (that always follows the advice), we see that the straight line tradeoff is attainable (by the same reasoning as the "coin-flip" algorithm in the paper).
> 3. To show the tradeoff is tight, consider a first-stage graph that is just a single edge with weight 1, and the advice suggests matching the edge. Let $x$ be the amount we match the edge. What is the maximum $x$ can be while still ensuring R-robustness? The answer is $x \leq 1-R$, because if $x > 1 - R$ then we cannot maintain R-robustness if in the second stage, a single edge to the offline vertex arrives with very high weight. Since $x \leq 1-R$, this means the maximum consistency on this instance is $1-R$ (which is the case if the second-stage graph is empty).
>
> Even though this setting is more general, we focused on the vertex-weighted setting in the paper because the tight tradeoff curve (as we show) is *better* than the straight-line interpolation, which we thought was more interesting.

---

> > ### Comment · Reviewer_d2AC · 2022-08-09
> > **Response**
> >
> > Thanks for the clear explanation.

---

### Official Review · Reviewer_CvJo · 2022-07-11

**Rating:** 7
**Confidence:** 4
**Soundness:** 4 excellent
**Presentation:** 4 excellent
**Contribution:** 3 good

**Summary:**

The paper studies online matching with advice. In particular, the authors focus on a two-stage online bipartite matching problem, where vertices of part A are weighted and known in advance, while vertices of part B arrive in two batches. Each vertex arrives with all its incident edges, and we need to make irrevocable decision about matching each vertex in B.

Similar to the new literature on advice-augmented algorithms, a (supposedly good) matching for the first batch is given before the second stage begins. Two measures of robustness and consistency are important in this regime. Robustness compares with the optimal solution regardless of the advice, and consistency looks at optimality assuming the advice is correct. Compared to the usual prediction-error bounds, consistency is simpler to understand (one can translate between them).

The authors manage to characterize the Pareto frontier of robustness and consistency by presenting an essentially optimal algorithm. Note that without advice, a 3/4-competitive algorithm exists (due to [FNS21]), which trivially translates to a (3/4, 3/4) advice-augmented solution. A naive solution following only the advice obtains a (0, 1) guarantee (0 for robustness and 1 for consistency). While with a coin flip between the two methods, one can get a smooth linear tradeoff, the authors show how to use different set of penalties to obtain the optimal tradeoff of $\sqrt{1-R} + \sqrt{1-C} = 1$.

**Questions:**

none

**Strengths And Weaknesses:**

Strengths:
-First characterization of Pareto frontier for online bipartite matching.
-Tight guarantees.

Weaknesses:
- Lack of experiments on actual online bipartite matching datasets.

---

> ### Author Response · Authors · 2022-08-01
> **Response to Reviewer CvJo**
>
> Thank you for the review!

---

### Official Review · Reviewer_Lbyq · 2022-07-12

**Rating:** 7
**Confidence:** 4
**Soundness:** 4 excellent
**Presentation:** 4 excellent
**Contribution:** 3 good

**Summary:**

The paper considers the vertex-weighted online bipartite matching problem where the online vertices arrive in two batches. In the first batch, along with the neighborhoods of all vertices of the first batch, the algorithm is also provided with a predicted matching. The algorithm needs to commit to a matching, and then the second batch of online vertices are revealed and the algorithm can now match these new vertices to the set of unmatched offline vertices. The goal of the algorithm is to maximize the weight of the final matching. The paper defines the “robustness” of the algorithm as the ratio of the weight of the matching found by the algorithm to the weight of the optimal offline matching, and the “consistency” as the ratio of the weight of the matching found by the algorithm to the weight of the best matching obtained by following the predicted matching exactly. The primary contribution of the paper is to quantify the exact trade-off between these quantities.


**Questions:**

None

**Strengths And Weaknesses:**

It’s great that the authors can obtain a tight and non-trivial tradeoff between robustness and consistency for this model. The paper is well-written and is easy to read and follow.

The analysis and algorithm are based on Feng et al, and the primary observation is that any penalty function within a range of functions is sufficient to guarantee the desired robustness guarantee. By then adjusting the penalties for vertices in the predicted matching and for those not in the predicted matching, the bound on consistency follows.

I appreciate the discussion in the introduction on alternate definitions of consistency and the notion of prediction error.

---

> ### Author Response · Authors · 2022-08-01
> **Response to Reviewer Lbyq**
>
> Thank you for the review!

---

### Meta-Review · Area_Chair_NN54 · 2022-08-25

**Recommendation:** Accept
**Confidence:** Certain

**Metareview:**

The paper studies two stage matching with advice and completely characterizes the tradeoff between the robustness and consistency. The online matching problem is a central problem in online algorithms with numerous applications such as assigning jobs to machines, impressions to advertisers, etc. The model studied here is a very simplified online model where there are only two stages. On the other hand, the paper is a rare case in advice augmented algorithms where the tradeoff between robustness and consistency is fully understood. The reviewers all appreciate the tight characterization. There is a minor concern that the paper does not include experimental evaluation. The authors are encouraged to include the simple result for the edge weighted case (in the author response) to provide more context to the main result.

**Award:**

No

---

### Decision · Program_Chairs · 2022-09-14

Accept